# Fine-grained Analysis of Sentence Embeddings Using Auxiliary Prediction Tasks

**Yossi Adi**[1,2]**, Einat Kermany**[2]**, Yonatan Belinkov**[3]**, Ofer Lavi**[2]**, Yoav Goldberg**[1]

[1]Bar-Ilan University, Ramat-Gan, Israel
{yoav.goldberg, yossiadidrum}@gmail.com
[2]IBM Haifa Research Lab, Haifa, Israel
{einatke, oferl}@il.ibm.com
[3]MIT Computer Science and Artificial Intelligence Laboratory, Cambridge, MA, USA
 belinkov@mit.edu

## Abstract

There is a lot of research interest in encoding variable length sentences into fixed length vectors, in a way that preserves the sentence meanings. Two common methods include representations based on averaging word vectors, and representations based on the hidden states of recurrent neural networks such as LSTMs. The sentence vectors are used as features for subsequent machine learning tasks or for pre-training in the context of deep learning. However, not much is known about the properties that are encoded in these sentence representations and about the language information they capture.

We propose a framework that facilitates better understanding of the encoded representations. We define prediction tasks around isolated aspects of sentence structure (namely sentence length, word content, and word order), and score representations by the ability to train a classifier to solve each prediction task when using the representation as input. We demonstrate the potential contribution of the approach by analyzing different sentence representation mechanisms. The analysis sheds light on the relative strengths of different sentence embedding methods with respect to these low level prediction tasks, and on the effect of the encoded vector's dimensionality on the resulting representations.

## 1 Introduction

While *sentence embeddings* or *sentence representations* play a central role in recent deep learning approaches to NLP, little is known about the information that is captured by different sentence embedding learning mechanisms. We propose a methodology facilitating fine-grained measurement of some of the information encoded in sentence embeddings, as well as performing fine-grained comparison of different sentence embedding methods.

In sentence embeddings, sentences, which are variable-length sequences of discrete symbols, are encoded into fixed length continuous vectors that are then used for further prediction tasks. A simple and common approach is producing word-level vectors using, e.g., word2vec (Mikolov et al., 2013a;b), and summing or averaging the vectors of the words participating in the sentence. This continuous-bag-of-words (CBOW) approach disregards the word order in the sentence.[1]

Another approach is the *encoder-decoder* architecture, producing models also known as *sequence-to-sequence* models (Sutskever et al., 2014; Cho et al., 2014; Bahdanau et al., 2014, inter alia). In this architecture, an *encoder* network (e.g. an LSTM) is used to produce a vector representation of the sentence, which is then fed as input into a *decoder* network that uses it to perform some prediction task (e.g. recreate the sentence, or produce a translation of it). The encoder and decoder networks are trained jointly in order to perform the final task.

---

[1]We use the term CBOW to refer to a sentence representation that is composed of an average of the vectors of the words in the sentence, not to be confused with the training method by the same name which is used in the word2vec algorithm.

Some systems (for example in machine translation) train the system end-to-end, and use the trained system for prediction (Bahdanau et al., 2014). Such systems do not generally care about the encoded vectors, which are used merely as intermediate values. However, another common case is to train an encoder-decoder network and then throw away the decoder and use the trained encoder as a general mechanism for obtaining sentence representations. For example, an encoder-decoder network can be trained as an auto-encoder, where the encoder creates a vector representation, and the decoder attempts to recreate the original sentence (Li et al., 2015). Similarly, Kiros et al. (2015) train a network to encode a sentence such that the decoder can recreate its neighboring sentences in the text. Such networks do not require specially labeled data, and can be trained on large amounts of unannotated text. As the decoder needs information about the sentence in order to perform well, it is clear that the encoded vectors capture a non-trivial amount of information about the sentence, making the encoder appealing to use as a general purpose, stand-alone sentence encoding mechanism. The sentence encodings can then be used as input for other prediction tasks for which less training data is available (Dai & Le, 2015). In this work we focus on these "general purpose" sentence encodings.

The resulting sentence representations are opaque, and there is currently no good way of comparing different representations short of using them as input for different high-level semantic tasks (e.g. sentiment classification, entailment recognition, document retrieval, question answering, sentence similarity, etc.) and measuring how well they perform on these tasks. This is the approach taken by Li et al. (2015), Hill et al. (2016) and Kiros et al. (2015). This method of comparing sentence embeddings leaves a lot to be desired: the comparison is at a very coarse-grained level, does not tell us much about the kind of information that is encoded in the representation, and does not help us form generalizable conclusions.

**Our Contribution**   We take a first step towards opening the black box of vector embeddings for sentences. We propose a methodology that facilitates comparing sentence embeddings on a much finer-grained level, and demonstrate its use by analyzing and comparing different sentence representations. We analyze sentence representation methods that are based on LSTM auto-encoders and the simple CBOW representation produced by averaging word2vec word embeddings. For each of CBOW and LSTM auto-encoder, we compare different numbers of dimensions, exploring the effect of the dimensionality on the resulting representation. We also provide some comparison to the skip-thought embeddings of Kiros et al. (2015).

In this work, we focus on what are arguably the three most basic characteristics of a sequence: its length, the items within it, and their order. We investigate different sentence representations based on the capacity to which they encode these aspects. Our analysis of these low-level properties leads to interesting, actionable insights, exposing relative strengths and weaknesses of the different representations.

**Limitations**   Focusing on low-level sentence properties also has limitations: The tasks focus on measuring the preservation of surface aspects of the sentence and do not measure syntactic and semantic generalization abilities; the tasks are not directly related to any specific downstream application (although the properties we test are important factors in many tasks – knowing that a model is good at predicting length and word order is likely advantageous for syntactic parsing, while models that excel at word content are good for text classification tasks). Dealing with these limitations requires a complementary set of auxiliary tasks, which is outside the scope of this study and is left for future work.

The study also suffers from the general limitations of empirical work: we do not prove general theorems but rather measure behaviors on several data points and attempt to draw conclusions from these measurements. There is always the risk that our conclusions only hold for the datasets on which we measured, and will not generalize. However, we do consider our large sample of sentences from Wikipedia to be representative of the English language, at least in terms of the three basic sentence properties that we study.

**Summary of Findings**   Our analysis reveals the following insights regarding the different sentence embedding methods:

- Sentence representations based on averaged word vectors are surprisingly effective, and encode a non-trivial amount of information regarding sentence length. The information they contain

can also be used to reconstruct a non-trivial amount of the original word order in a probabilistic manner (due to regularities in the natural language data).

- LSTM auto-encoders are very effective at encoding word order and word content.

- Increasing the number of dimensions benefits some tasks more than others.

- Adding more hidden units sometimes *degrades* the encoders' ability to encode word content. This degradation is *not correlated* with the BLEU scores of the decoder, suggesting that BLEU over the decoder output is sub-optimal for evaluating the encoders' quality.

- LSTM encoders trained as auto-encoders do not rely on ordering patterns in the training sentences when encoding novel sentences, while the skip-thought encoders do rely on such patterns.

## 2 RELATED WORK

Word-level distributed representations have been analyzed rather extensively, both empirically and theoretically, for example by Baroni et al. (2014), Levy & Goldberg (2014) and Levy et al. (2015). In contrast, the analysis of sentence-level representations has been much more limited. Commonly used approaches is to either compare the performance of the sentence embeddings on down-stream tasks (Hill et al., 2016), or to analyze models, specifically trained for predefined task (Schmaltz et al., 2016; Sutskever et al., 2011).

While the resulting analysis reveals differences in performance of different models, it does not adequately explain what kind of linguistic properties of the sentence they capture. Other studies analyze the hidden units learned by neural networks when training a sentence representation model (Elman, 1991; Karpathy et al., 2015; Kádár et al., 2016). This approach often associates certain linguistic aspects with certain hidden units. Kádár et al. (2016) propose a methodology for quantifying the contribution of each input word to a resulting GRU-based encoding. These methods depend on the specific learning model and cannot be applied to arbitrary representations. Moreover, it is still not clear what is captured by the final sentence embeddings.

Our work is orthogonal and complementary to the previous efforts: we analyze the resulting sentence embeddings by devising auxiliary prediction tasks for core sentence properties. The methodology we purpose is general and can be applied to any sentence representation model.

## 3 APPROACH

We aim to inspect and compare encoded sentence vectors in a task-independent manner. The main idea of our method is to focus on isolated aspects of sentence structure, and design experiments to measure to what extent each aspect is captured in a given representation.

In each experiment, we formulate a prediction task. Given a sentence representation method, we create training data and train a classifier to predict a specific sentence property (e.g. their length) based on their vector representations. We then measure how well we can train a model to perform the task. The basic premise is that if we cannot train a classifier to predict some property of a sentence based on its vector representation, then this property is not encoded in the representation (or rather, not encoded in a useful way, considering how the representation is likely to be used).

The experiments in this work focus on low-level properties of sentences – the sentence length, the identities of words in a sentence, and the order of the words. We consider these to be the core elements of sentence structure. Generalizing the approach to higher-level semantic and syntactic properties holds great potential, which we hope will be explored in future work, by us or by others.

### 3.1 THE PREDICTION TASKS

We now turn to describe the specific prediction tasks. We use lower case italics ($s$, $w$) to refer to sentences and words, and boldface to refer to their corresponding vector representations ($\mathbf{s}$, $\mathbf{w}$). When more than one element is considered, they are distinguished by indices ($w_1$, $w_2$, $\mathbf{w_1}$, $\mathbf{w_2}$).

Our underlying corpus for generating the classification instances consists of 200,000 Wikipedia sentences, where 150,000 sentences are used to generate training examples, and 25,000 sentences

are used for each of the test and development examples. These sentences are a subset of the training set that was used to train the original sentence encoders. The idea behind this setup is to test the models on what are presumably their best embeddings.

**Length Task**   This task measures to what extent the sentence representation encodes its length. Given a sentence representation $\mathbf{s} \in \mathbb{R}^k$, the goal of the classifier is to predict the length (number of words) in the original sentence $s$. The task is formulated as multiclass classification, with eight output classes corresponding to binned lengths.[2] The resulting dataset is reasonably balanced, with a majority class (lengths 5-8 words) of 5,182 test instances and a minority class (34-70) of 1,084 test instances. Predicting the majority class results in classification accuracy of 20.1%.

**Word-content Task**   This task measures to what extent the sentence representation encodes the identities of words within it. Given a sentence representation $\mathbf{s} \in \mathbb{R}^k$ and a word representation $\mathbf{w} \in \mathbb{R}^d$, the goal of the classifier is to determine whether $w$ appears in the $s$, with access to neither $w$ nor $s$. This is formulated as a binary classification task, where the input is the concatenation of $\mathbf{s}$ and $\mathbf{w}$.

To create a dataset for this task, we need to provide positive and negative examples. Obtaining positive examples is straightforward: we simply pick a random word from each sentence. For negative examples, we could pick a random word from the entire corpus. However, we found that such a dataset tends to push models to memorize words as either positive or negative words, instead of finding their relation to the sentence representation. Therefore, for each sentence we pick as a negative example a word that appears as a positive example somewhere in our dataset, but does not appear in the given sentence. This forces the models to learn a relationship between word and sentence representations. We generate one positive and one negative example from each sentence. The dataset is balanced, with a baseline accuracy of 50%.

**Word-order Task**   This task measures to what extent the sentence representation encodes word order. Given a sentence representation $\mathbf{s} \in \mathbb{R}^k$ and the representations of two words that appear in the sentence, $\mathbf{w}_1, \mathbf{w}_2 \in \mathbb{R}^d$, the goal of the classifier is to predict whether $w_1$ appears before or after $w_2$ in the original sentence $s$. Again, the model has no access to the original sentence and the two words. This is formulated as a binary classification task, where the input is a concatenation of the three vectors $\mathbf{s}$, $\mathbf{w}_1$ and $\mathbf{w}_2$.

For each sentence in the corpus, we simply pick two random words from the sentence as a positive example. For negative examples, we flip the order of the words. We generate one positive and one negative example from each sentence. The dataset is balanced, with a baseline accuracy of 50%.

## 4   SENTENCE REPRESENTATION MODELS

Given a sentence $s = \{w_1, w_2, ..., w_N\}$ we aim to find a sentence representation $\mathbf{s}$ using an encoder:

$$\text{ENC} : s = \{w_1, w_2, ..., w_N\} \mapsto \mathbf{s} \in \mathbb{R}^k$$

The encoding process usually assumes a vector representation $\mathbf{w}_i \in \mathbb{R}^d$ for each word in the vocabulary. In general, the word and sentence embedding dimensions, $d$ and $k$, need not be the same. The word vectors can be learned together with other encoder parameters or pre-trained. Below we describe different instantiations of ENC.

**Continuous Bag-of-words (CBOW)**   This simple yet effective text representation consists of performing element-wise averaging of word vectors that are obtained using a word-embedding method such as word2vec.

Despite its obliviousness to word order, CBOW has proven useful in different tasks (Hill et al., 2016) and is easy to compute, making it an important model class to consider.

**Encoder-Decoder (ED)**   The encoder-decoder framework has been successfully used in a number of sequence-to-sequence learning tasks (Sutskever et al., 2014; Bahdanau et al., 2014; Dai & Le, 2015; Li et al., 2015). After the encoding phase, a decoder maps the sentence representation back to the sequence of words:

$$\text{DEC} : \mathbf{s} \in \mathbb{R}^k \mapsto s = \{w_1, w_2, ..., w_N\}$$

---

[2] We use the bins (5-8), (9-12), (13-16), (17-20), (21-25), (26-29), (30-33), (34-70).

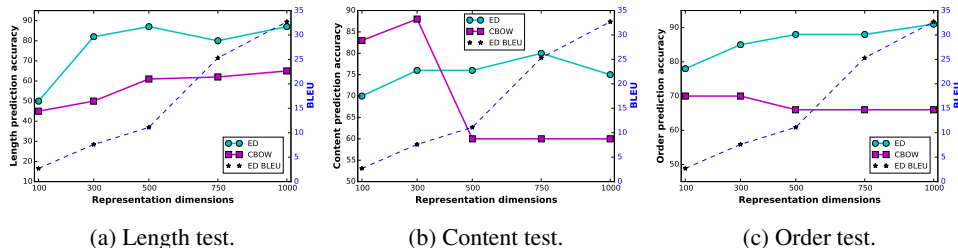

| (a) Length test. | (b) Content test. | (c) Order test. |

Figure 1: Task accuracy vs. embedding size for different models; ED BLEU scores given for reference.

Here we investigate the specific case of an auto-encoder, where the entire encoding-decoding process can be trained end-to-end from a corpus of raw texts. The sentence representation is the final output vector of the encoder. We use a long short-term memory (LSTM) recurrent neural network (Hochreiter & Schmidhuber, 1997; Graves et al., 2013) for both encoder and decoder. The LSTM decoder is similar to the LSTM encoder but with different weights.

## 5 Experimental Setup

The bag-of-words (CBOW) and encoder-decoder models are trained on 1 million sentences from a 2012 Wikipedia dump with vocabulary size of 50,000 tokens. We use NLTK (Bird, 2006) for tokenization, and constrain sentence lengths to be between 5 and 70 words. For both models we control the embedding size $k$ and train word and sentence vectors of sizes $k \in \{100, 300, 500, 750, 1000\}$. More details about the experimental setup are available in the Appendix.

## 6 Results

In this section we provide a detailed description of our experimental results along with their analysis. For each of the three main tests – length, content and order – we investigate the performance of different sentence representation models across embedding size.

### 6.1 Length Experiments

We begin by investigating how well the different representations encode sentence length. Figure 1a shows the performance of the different models on the length task, as well as the BLEU obtained by the LSTM encoder-decoder (ED).

With enough dimensions, the LSTM embeddings are very good at capturing sentence length, obtaining accuracies between 82% and 87%. Length prediction ability is not perfectly correlated with BLEU scores: from 300 dimensions onward the length prediction accuracies of the LSTM remain relatively stable, while the BLEU score of the encoder-decoder model increases as more dimensions are added.

Somewhat surprisingly, the CBOW model also encodes a fair amount of length information, with length prediction accuracies of 45% to 65%, way above the 20% baseline. This is remarkable, as the CBOW representation consists of averaged word vectors, and we did not expect it to encode length at all. We return to CBOW's exceptional performance in Section 7.

### 6.2 Word Content Experiments

To what extent do the different sentence representations encode the identities of the words in the sentence? Figure 1b visualizes the performance of our models on the word content test.

All the representations encode some amount of word information, and clearly outperform the random baseline of 50%. Some trends are worth noting. While the capacity of the LSTM encoder to preserve word identities generally increases when adding dimensions, the performance peaks at 750 dimensions and drops afterwards. This stands in contrast to the BLEU score of the respective

encoder-decoder models. We hypothesize that this occurs because a sizable part of the auto-encoder performance comes from the decoder, which also improves as we add more dimensions. At 1000 dimensions, the decoder's language model may be strong enough to allow the representation produced by the encoder to be less informative with regard to word content.

CBOW representations with low dimensional vectors (100 and 300 dimensions) perform exceptionally well, outperforming the more complex, sequence-aware models by a wide margin. If your task requires access to word identities, it is worth considering this simple representation. Interestingly, CBOW scores drop at higher dimensions.

### 6.3 WORD ORDER EXPERIMENTS

Figure 1c shows the performance of the different models on the order test. The LSTM encoders are very capable of encoding word order, with LSTM-1000 allowing the recovery of word order in 91% of the cases. Similar to the length test, LSTM order prediction accuracy is only loosely correlated with BLEU scores. It is worth noting that increasing the representation size helps the LSTM-encoder to better encode order information.

Surprisingly, the CBOW encodings manage to reach an accuracy of 70% on the word order task, 20% above the baseline. This is remarkable as, by definition, the CBOW encoder does not attempt to preserve word order information. One way to explain this is by considering distribution patterns of words in natural language sentences: some words tend to appear before others. In the next section we analyze the effect of natural language on the different models.

## 7 IMPORTANCE OF "NATURAL LANGUAGENESS"

Natural language imposes many constraints on sentence structure. To what extent do the different encoders rely on specific properties of word distributions in natural language sentences when encoding sentences?

To account for this, we perform additional experiments in which we attempt to control for the effect of natural language.

**How can CBOW encode sentence length?**   Is the ability of CBOW embeddings to encode length related to specific words being indicative of longer or shorter sentences? To control for this, we created a synthetic dataset where each word in each sentence is replaced by a random word from the dictionary and re-ran the length test for the CBOW embeddings using this dataset. As Figure 2a shows, this only leads to a slight decrease in accuracy, indicating that the identity of the words is not the main component in CBOW's success at predicting length.

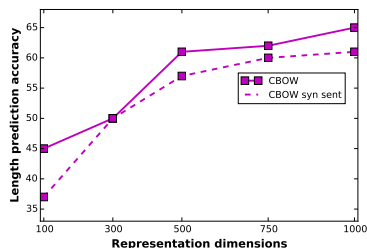
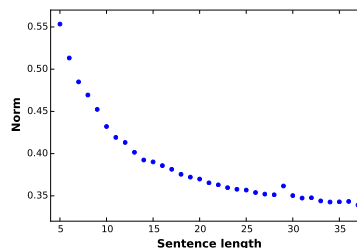

(a) Length accuracy for different CBOW sizes on natural and synthetic (random words) sentences.

(b) Average embedding norm vs. sentence length for CBOW with an embedding size of 300.

An alternative explanation for CBOW's ability to encode sentence length is given by considering the norms of the sentence embeddings. Indeed, Figure 2b shows that the embedding norm decreases as sentences grow longer. We believe this is one of the main reasons for the strong CBOW results.

While the correlation between the number of averaged vectors and the resulting norm surprised us, in retrospect it is an expected behavior that has sound mathematical foundations. To understand the behavior, consider the different word vectors to be random variables, with the values in each

dimension centered roughly around zero. Both central limit theorem and Hoeffding's inequality tell us that as we add more samples, the expected average of the values will better approximate the true mean, causing the norm of the average vector to decrease. We expect the correlation between the sentence length and its norm to be more pronounced with shorter sentences (above some number of samples we will already be very close to the true mean, and the norm will not decrease further), a behavior which we indeed observe in practice.

**How does CBOW encode word order?** The surprisingly strong performance of the CBOW model on the order task made us hypothesize that much of the word order information is captured in general natural language word order statistics.

To investigate this, we re-run the word order tests, but this time drop the sentence embedding in training and testing time, learning from the word-pairs alone. In other words, we feed the network as input two word embeddings and ask which word comes first in the sentence. This test isolates general word order statistics of language from information that is contained in the sentence embedding (Fig. 3).

The difference between including and removing the sentence embeddings when using the CBOW model is minor, while the LSTM-ED suffers a significant drop. Clearly, the LSTM-ED model encodes word order, while the prediction ability of CBOW is mostly explained by general language statistics. However, CBOW does benefit from the sentence to some extent: we observe a gain of ∼3% accuracy points when the CBOW tests are allowed access to the sentence representation. This may be explained by higher order statistics of correlation between word order patterns and the occurrences of specific words.

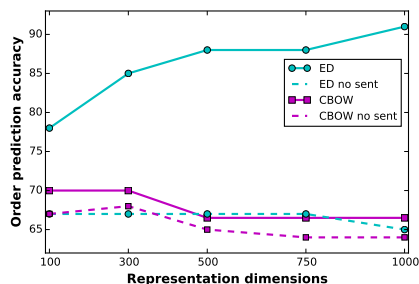

Figure 3: Order accuracy w/ and w/o sentence representation for ED and CBOW models.

**How important is English word order for encoding sentences?** To what extent are the models trained to rely on natural language word order when encoding sentences? To control for this, we create a synthetic dataset, PERMUTED, in which the word order in each sentence is randomly permuted. Then, we repeat the length, content and order experiments using the PERMUTED dataset (we still use the original sentence encoders that are trained on non-permuted sentences). While the permuted sentence representation is the same for CBOW, it is completely different when generated by the encoder-decoder.

Results are presented in Fig. 4. When considering CBOW embeddings, word order accuracy drops to chance level, as expected, while results on the other tests remain the same. Moving to the LSTM encoder-decoder, the results on all three tests are comparable to the ones using non-permuted sentences. These results are somewhat surprising since the models were originally trained on "real", non-permuted sentences. This indicates that the LSTM encoder-decoder is a general-purpose sequence encoder that for the most part does not rely on word ordering properties of natural language when encoding sentences. The small and consistent drop in word order accuracy on the permuted sentences can be attributed to the encoder relying on natural language word order to some extent, but can also be explained by the word order prediction task becoming harder due to the inability to

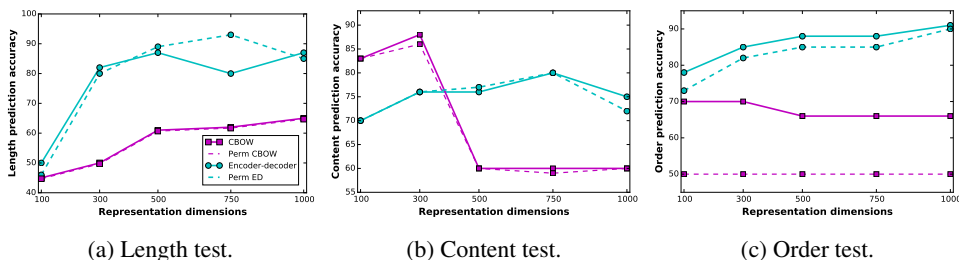

| (a) Length test. | (b) Content test. | (c) Order test. |

Figure 4: Results for length, content and order tests on natural and permuted sentences.

use general word order statistics. The results suggest that a trained encoder will transfer well across different natural language domains, as long as the vocabularies remain stable. When considering the decoder's BLEU score on the permuted dataset (not shown), we do see a dramatic decrease in accuracy. For example, LSTM encoder-decoder with 1000 dimensions drops from 32.5 to 8.2 BLEU score. These results suggest that the decoder, which is thrown away, contains most of the language-specific information.

## 8    SKIP-THOUGHT VECTORS

In addition to the experiments on CBOW and LSTM-encoders, we also experiment with the skip-thought vectors model (Kiros et al., 2015). This model extends the idea of the auto-encoder to neighboring sentences.

Given a sentence $s_i$, it first encodes it using an RNN, similar to the auto-encoder model. However, instead of predicting the original sentence, skip-thought predicts the preceding and following sentences, $s_{i-1}$ and $s_{i+1}$. The encoder and decoder are implemented with gated recurrent units (Cho et al., 2014).

Here, we deviate from the controlled environment and use the author's provided model[3] with the recommended embeddings size of 4800. This makes the direct comparison of the models "unfair". However, our aim is not to decide which is the "best" model but rather to show how our method can be used to measure the kinds of information captured by different representations.

Table 1 summarizes the performance of the skip-thought embeddings in each of the prediction tasks on both the PERMUTED and original dataset.

|  | Length | Word content | Word order |
|---|---|---|---|
| **Original** | 82.1% | 79.7% | 81.1% |
| **Permuted** | 68.2% | 76.4% | 76.5% |

Table 1: Classification accuracy for the prediction tasks using skip-thought embeddings.

The performance of the skip-thought embeddings is well above the baselines and roughly similar for all tasks. Its performance is similar to the higher-dimensional encoder-decoder models, except in the order task where it lags somewhat behind. However, we note that the results are not directly comparable as skip-thought was trained on a different corpus.

The more interesting finding is its performance on the PERMUTED sentences. In this setting we see a large drop. In contrast to the LSTM encoder-decoder, skip-thought's ability to predict length and word content does degrade significantly on the permuted sentences, suggesting that the encoding process of the skip-thought model is indeed specialized towards natural language texts.

## 9    CONCLUSION

We presented a methodology for performing fine-grained analysis of sentence embeddings using auxiliary prediction tasks. Our analysis reveals some properties of sentence embedding methods:

- CBOW is surprisingly effective – in addition to being very strong at content, it is *also predictive of length, and can be used to reconstruct a non-trivial amount of the original word order*. 300 dimensions perform best, with greatly degraded word-content prediction performance on higher dimensions.

- With enough dimensions, LSTM auto-encoders are very effective at encoding word order and word content information. Increasing the dimensionality of the LSTM encoder does not significantly improve its ability to encode length, but does increase its ability to encode content and order information. 500 dimensional embeddings are already quite effective for encoding word order, with little gains beyond that. Word content accuracy peaks at 750 dimensions and drops at 1000, suggesting that *larger is not always better*.

---

[3]https://github.com/ryankiros/skip-thoughts

- The trained LSTM encoder (when trained with an auto-encoder objective) *does not rely* on ordering patterns in the training sentences when encoding novel sequences.

  In contrast, the skip-thought encoder *does rely* on such patterns. Its performance on the other tasks is similar to the higher-dimensional LSTM encoder, which is impressive considering it was trained on a different corpus.

- Finally, the encoder-decoder's ability to recreate sentences (BLEU) is not entirely indicative of the quality of the encoder at representing aspects such as word identity and order. This suggests that *BLEU is sub-optimal for model selection*.

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

## APPENDIX I: EXPERIMENTAL SETUP

**Sentence Encoders**  The bag-of-words (CBOW) and encoder-decoder models are trained on 1 million sentences from a 2012 Wikipedia dump with vocabulary size of 50,000 tokens. We use NLTK (Bird, 2006) for tokenization, and constrain sentence lengths to be between 5 and 70 words.

For the CBOW model, we train Skip-gram word vectors (Mikolov et al., 2013a), with hierarchical-softmax and a window size of 5 words, using the Gensim implementation.[4] We control for the embedding size $k$ and train word vectors of sizes $k \in \{100, 300, 500, 750, 1000\}$.

For the encoder-decoder models, we use an in-house implementation using the Torch7 toolkit (Collobert et al., 2011). The decoder is trained as a language model, attempting to predict the correct word at each time step using a negative-log-likelihood objective (cross-entropy loss over the softmax layer). We use one layer of LSTM cells for the encoder and decoder using the implementation in Léonard et al. (2015).

We use the same size for word and sentence representations (i.e. $d = k$), and train models of sizes $k \in \{100, 300, 500, 750, 1000\}$. We follow previous work on sequence-to-sequence learning (Sutskever et al., 2014; Li et al., 2015) in reversing the input sentences and clipping gradients. Word vectors are initialized to random values.

We evaluate the encoder-decoder models using BLEU scores (Papineni et al., 2002), a popular machine translation evaluation metric that is also used to evaluate auto-encoder models (Li et al., 2015). BLEU score measures how well the original sentence is recreated, and can be thought of as a proxy for the quality of the encoded representation. We compare it with the performance of the models on the three prediction tasks. The results of the higher-dimensional models are comparable to those found in the literature, which serves as a sanity check for the quality of the learned models.

**Auxiliary Task Classifier**  For the auxiliary task predictors, we use multi-layer perceptrons with a single hidden layer and ReLU activation, which were carefully tuned for each of the tasks. We experimented with several network architectures prior to arriving at this configuration.

Further details regarding the training and architectures of both the sentence encoders and auxiliary task classifiers are available in the Appendix.

## APPENDIX II: TECHNICAL DETAILS

### ENCODER DECODER

Parameters of the encoder-decoder were tuned on a dedicated validation set. We experienced with different learning rates (0.1, 0.01, 0.001), dropout-rates (0.1, 0.2, 0.3, 0.5) (Hinton et al., 2012) and optimization techniques (AdaGrad (Duchi et al., 2011), AdaDelta (Zeiler, 2012), Adam (Kingma & Ba, 2014) and RMSprop (Tieleman & Hinton, 2012)). We also experimented with different batch sizes (8, 16, 32), and found improvement in runtime but no significant improvement in performance.

Based on the tuned parameters, we trained the encoder-decoder models on a single GPU (NVIDIA Tesla K40), with mini-batches of 32 sentences, learning rate of 0.01, dropout rate of 0.1, and the AdaGrad optimizer; training takes approximately 10 days and is stopped after 5 epochs with no loss improvement on a validation set.

### PREDICTION TASKS

Parameters for the predictions tasks as well as classifier architecture were tuned on a dedicated validation set. We experimented with one, two and three layer feed-forward networks using ReLU (Nair & Hinton, 2010; Glorot et al., 2011), tanh and sigmoid activation functions. We tried different hidden layer sizes: the same as the input size, twice the input size and one and a half times the input size. We tried different learning rates (0.1, 0.01, 0.001), dropout rates (0.1, 0.3, 0.5, 0.8) and different optimization techniques (AdaGrad, AdaDelta and Adam).

---

[4]https://radimrehurek.com/gensim

Our best tuned classifier, which we use for all experiments, is a feed-forward network with one hidden layer and a ReLU activation function. We set the size of the hidden layer to be the same size as the input vector. We place a softmax layer on top whose size varies according to the specific task, and apply dropout before the softmax layer. We optimize the log-likelihood using AdaGrad. We use a dropout rate of 0.8 and a learning rate of 0.01. Training is stopped after 5 epochs with no loss improvement on the development set. Training was done on a single GPU (NVIDIA Tesla K40).

## 10 ADDITIONAL EXPERIMENTS - CONTENT TASK

How well do the models preserve content when we increase the sentence length? In Fig. 5 we plot content prediction accuracy vs. sentence length for different models.

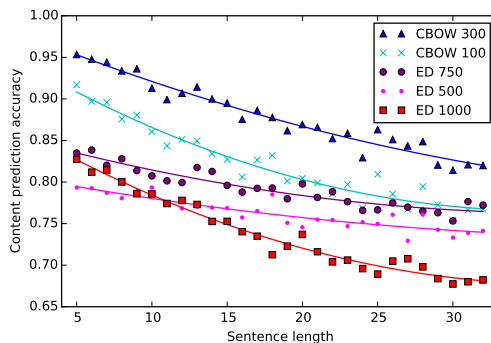

Figure 5: Content accuracy vs. sentence length for selected models.

As expected, all models suffer a drop in content accuracy on longer sentences. The degradation is roughly linear in the sentence length. For the encoder-decoder, models with fewer dimensions seem to degrade slower.

## APPENDIX III: SIGNIFICANCE TESTS

In this section we report the significance tests we conduct in order to evaluate our findings. In order to do so, we use the paired t-test (Rubin, 1973).

All the results reported in the summery of findings are highly significant (p-value $\ll 0.0001$). The ones we found to be not significant (p-value $\gg 0.03$) are the ones which their accuracy does not have much of a difference, i.e ED with size 500 and ED with size 750 tested on the word order task (p-value=0.11), or CBOW with dimensions 750 and 1000 (p-value=0.3).

| Dim. | Length | Word content | Word order |
|---|---|---|---|
| **100** | 1.77e-147 | 0.0 | 1.83e-296 |
| **300** | 0.0 | 0.0 | 0.0 |
| **500** | 0.0 | 0.0 | 0.0 |
| **750** | 0.0 | 0.0 | 0.0 |
| **1000** | 0.0 | 0.0 | 0.0 |

Table 2: P-values for ED vs. CBOW over the different dimensions and tasks. For example, in the row where dim equals 100, we compute the p-value of ED compared to CBOW with embed size of 100 on all three tasks.

| Dim. | Length | Word content | Word order |
|---|---|---|---|
| **100 vs. 300** | 0.0 | 8.56e-190 | 0.0 |
| **300 vs. 500** | 7.3e-71 | 4.20e-05 | 5.48e-56 |
| **500 vs. 750** | 3.64e-175 | 4.46e-65 | **0.11** |
| **750 vs. 1000** | 1.37e-111 | 2.35e-243 | 4.32e-61 |

Table 3: P-values for ED models over the different dimensions and tasks.

| Dim. | Length | Word content | Word order |
|---|---|---|---|
| **100 vs. 300** | 0.0 | 0.0 | 1.5e-33 |
| **300 vs. 500** | 1.47e-215 | 0.0 | 3.06e-64 |
| **500 vs. 750** | **0.68** | **0.032** | **0.05** |
| **750 vs. 1000** | 4.44e-32 | **0.3** | **0.08** |

Table 4: P-values for CBOW models over the different dimensions and tasks.

