# Peer review of "Fine-grained Analysis of Sentence Embeddings Using Auxiliary Prediction Tasks"

_ICLR 2017 — accepted_

[Public Comment · Jiaqi Mu · 17 Nov 2016]
**applications of task (c)**

We (J. Mu and P. Viswanath) enjoyed the tour-de-force comparison of a vast variety of sentence representation algorithms all in one compact manuscript. Of particular interest to us were the three  "synthetic" tasks introduced here:  (a) to what extent the sentence representation encodes its length; (b) to what extent the sentence representation encodes the identities of words within it and (c) to what extent the sentence representation encodes word order.  

The best part of these tasks is that they are very well defined and labeling does not need any (expert) supervision at all and can be done over the entire corpus too. We have a reasonable intuition on why tasks (a) and (b) might be interesting/relevant for downstream tasks: the length of a sentence could be a proxy for the amount of content in the sentence; testing of a word within a sentence could be a type of test for the topic embedded in the sentence. 

But we aren't so clear as to what might be the sense in which task (c) could be useful for downstream applications. One instance where this matters seems to be cause-effect relationships. For example,  the order of 'Mary' and 'John' is critical in 'Mary stole an apple from John.' Such a pair of words tend to be  named entities, however. 

The tests presented in this manuscript worked with a random pair of words (and not just named entities or scenarios where cause-effect relationship mattered). We would love to hear what the authors think about the use cases of task (c) and our conjecture that they are particularly relevant in cause-effect scenarios.

[Official Review · AnonReviewer3 · rating 8 · confidence 4 · 11 Dec 2016]
**Interesting analytic results on unsupervised sentence encoders**

This paper presents a set of experiments investigating what kinds of information are captured in common unsupervised approaches to sentence representation learning. The results are non-trivial and somewhat surprising. For example, they show that it is possible to reconstruct word order from bag of words representations, and they show that LSTM sentence autoencoders encode interpretable features even for randomly permuted nonsense sentences.

Effective unsupervised sentence representation learning is an important and largely unsolved problem in NLP, and this kind of work seems like it should be straightforwardly helpful towards that end. In addition, the experimental paradigm presented here is likely more broadly applicable to a range of representation learning systems. Some of the results seem somewhat strange, but I see no major technical concerns, and think that that they are informative. I recommend acceptance.

One minor red flag: 
- The massive drop in CBOW performance in Figures 1b and 4b are not explained, and seem implausible enough to warrant serious further investigation. Can you be absolutely certain that those results would appear with a different codebase and different random seed implementing the same model? Fortunately, this point is largely orthogonal to the major results of the paper.

Two writing comments:
- I agree that the results with word order and CBOW are surprising, but I think it's slightly misleading to say that CBOW is predictive of word order. It doesn't represent word order at all, but it's possible to probabilistically reconstruct word order from the information that it does encode.
- Saying that "LSTM auto-encoders are more effective at encoding word order than word content" doesn't really make sense. These two quantities aren't comparable.

[Official Review · AnonReviewer1 · rating 8 · confidence 4 · 16 Dec 2016]
**Experimental analysis of unsupervised sentence embeddings**

This paper analyzes various unsupervised sentence embedding approaches by means of a set of auxiliary prediction tasks. By examining how well classifiers can predict word order, word content, and sentence length, the authors aim to assess how much and what type of information is captured by the different embedding models. The main focus is on a comparison between and encoder-decoder model (ED) and a permutation-invariant model, CBOW. (There is also an analysis of skip-thought vectors, but since it was trained on a different corpus it is hard to compare).

There are several interesting and perhaps counter-intuitive results that emerge from this analysis and the authors do a nice job of examining those results and, for the most part, explaining them. However, I found the discussion of the word-order experiment rather unsatisfying. It seems to me that the appropriate question should have been something like, 'How well does model X do compared to the theoretical upper bound which can be deduced from natural language statistics?' This is investigated from one angle in Section 7, but I would have preferred to the effect of natural language statistics discussed up front rather than presented as the explanation to a 'surprising' observation. I had a similar reaction to the word-order experiments.

Most of the interesting results, in my opinion, are about the ED model. It is fascinating that the LSTM encoder does not seem to rely on natural-language ordering statistics -- it seems like doing so should be a big win in terms of per-parameter expressivity. I also think that it's strange that word content accuracy begins to drop for high-dimensional embeddings. I suppose this could be investigated by handicapping the decoder.

Overall, this is a very nice paper investigating some aspects of the information content stored in various types of sentence embeddings. I recommend acceptance.

[Official Review · AnonReviewer2 · rating 8 · confidence 5 · 20 Dec 2016]

The authors present a methodology for analyzing sentence embedding techniques by checking how much the embeddings preserve information about sentence length, word content, and word order. They examine several popular embedding methods including autoencoding LSTMs, averaged word vectors, and skip-thought vectors. The experiments are thorough and provide interesting insights into the representational power of common sentence embedding strategies, such as the fact that word ordering is surprisingly low-entropy conditioned on word content.

Exploring what sort of information is encoded in representation learning methods for NLP is an important and under-researched area. For example, the tide of word-embeddings research was mostly stemmed after a thread of careful experimental results showing most embeddings to be essentially equivalent, culminating in "Improving Distributional Similarity with Lessons Learned from Word Embeddings" by Levy, Goldberg, and Dagan. As representation learning becomes even more important in NLP this sort of research will be even more important.

While this paper makes a valuable contribution in setting out and exploring a methodology for evaluating sentence embeddings, the evaluations themselves are quite simple and do not necessarily correlate with real-world desiderata for sentence embeddings (as the authors note in other comments, performance on these tasks is not a normative measure of embedding quality). For example, as the authors note, the ability of the averaged vector to encode sentence length is trivially to be expected given the central limit theorem (or more accurately, concentration inequalities like Hoeffding's inequality).

The word-order experiments were interesting. A relevant citation for this sort of conditional ordering procedure is "Generating Text with Recurrent Neural Networks" by Sutskever, Martens, and Hinton, who refer to the conversion of a bag of words into a sentence as "debagging."

Although this is just a first step in better understanding of sentence embeddings, it is an important one and I recommend this paper for publication.

[Final Decision · Program Chairs · 06 Feb 2017]
**ICLR committee final decision**

The area chair agrees with the reviewers and think this paper would be of interest to the ICLR audience. There is clearly more to be done in this area, but the authors do a good job shedding some light on what sentence embeddings can encode. We need more work like this that helps us understand what neural networks can model.